# The contrasting effects of body image and self-esteem in the makeup usage

**Anthonieta Looman Mafra** [ID]*, **Caio S. A. Silva, Marco A. C. Varella, Jaroslava V. Valentova** [ID]

Department of Experimental Psychology, University of São Paulo, São Paulo, Brazil

* looman.anthonieta@gmail.com

**Data Availability Statement:** Data used to this paper can be found by accessing the following link: https://osf.io/d2z7e/ (DOI:10.17605/OSF.IO/D2Z7E).

## Abstract

Women wearing makeup are perceived by others as more attractive, competent, dominant, and more socially prestigious. Individuals differ in how much and how frequently they use makeup. Some studies show that women with lower self-esteem use more makeup, probably to hide imperfections. However, women with higher self-esteem can also use makeup to attract attention. This study verified whether social and general self-esteem and body image are associated with makeup usage in Brazilian women. We collected data from 1,483 women ($M_{age}$ = 31.08; $SD$ = 11.15) about body image (appearance orientation and appearance evaluation), social self-esteem, general self-esteem, and makeup usage (frequency of makeup usage, time spent applying makeup per day, and money spent on makeup per month). Appearance orientation positively predicted frequency of makeup usage, time spent applying makeup, and money spent on makeup, whereas appearance evaluation inversely predicted money spent on makeup per month. Social self-esteem and general self-esteem also positively predicted money spent on makeup, but in different directions. The results suggest that the significance given to appearance and social interactions are importantly associated with makeup usage in women.

## Introduction

Surveys with American women showed that 78% spent one hour per day on their appearance (e.g., hair treatments, dressing up, and makeup). Taking on average 55 minutes of women's day, hair and makeup seemed to need more time invested than other appearance related behaviors. [1] American women did not only spend time on active appearance enhancing behaviors, but between the most watched categories by women on YouTube, the top two are appearance related [1]. Another study showed that American women spent, on average, 10 minutes on makeup in the morning and 85% tended to apply at least 16 products on their faces before leaving home. The results also concluded that New York women spent around 300,000 US dollars during their lifetime on facial cosmetics [2]. These examples illustrate the importance American women attribute to physical appearance and self-care [e.g., 3].

Women's attractiveness is an important factor in their lives, affecting how they feel about themselves. Several studies found a positive relationship between attractiveness and self-esteem

**Funding:** AM was supported by the FAPESP (Fundação de amparo à pesquisa do estado de são paulo, Grant number: 2018/16370-5), CS was supported by CNPq (Conselho nacional de desenvolvimento científico e tecnológico, Grant number 143811/2019-3), MV was supported by the CAPES (Coordenação de Aperfeiçoamento de Pessoal de Nível Superior, Grant number 33002010037P0—MEC/CAPES).

**Competing interests:** The authors have declared that no competing interests exist.

in women [e.g., 4, 5]. Cash et al. [6] found that physical appearance is positively correlated with appearance satisfaction, and Grilo et al. [7] found a positive relation between appearance evaluation and general self-esteem. However, these studies have focused on general self-esteem.

Whereas general self-esteem reflects how a person feels about themselves and their value in comparison to others [8], social self-esteem is how individuals feel about themselves during social interactions with others, and how those interactions affect their social value [9]. In this way, social self-esteem is more affected by social interactions than general self-esteem. Social self-esteem is positively linked to use of social network sites [10, 11] whereas general self-esteem appears to be inversely proportional to social network sites [12, 13], especially in women [14]. Vogel and colleagues [15] found that social comparison negatively affected general self-esteem negatively. Thus, the increase in digital media usage increased the pressure to look as good as possible because people were increasingly exposed to images and videos of very good looking individuals [16]. On the other hand, Steinsbekk and colleagues [14] found that self-oriented social network sites use was not related to general self-esteem. Therefore, social network sites might increase social self-esteem by promoting more social interactions.

Although appearance can be manipulated for better or for worse [17], the most common direction of appearance manipulation is appearance enhancement rather than worsening. Appearance enhancement is considered to be a self-promotion strategy in which one may capture more attention from others [18]. To enhance their attractiveness and to look better than same sex peers, women in Western societies use various tactics, such as high heels [e.g., 19], cosmetic surgeries [e.g., 20], and/or makeup [e.g., 21]. Interestingly, Kelley [16] interviewed 132 American college women and found that 37% of them reported they started using makeup because they were unsatisfied with their appearance. In sixth grade girls with negative self-esteem, using makeup increases pleasure pursuit [22]. Gentina et al. [23] found that makeup can also serve as a ritual of transition to adulthood among adolescents.

A recent experimental study compared women's self-perception without makeup and with professionally applied makeup and showed that made up women considered themselves more feminine, attractive, more satisfied with appearance, and as having higher self-esteem [24]. Similarly, women wearing their usual facial cosmetics rated themselves as more attractive than when the cosmetics were removed [6]. A cross-sectional study further reported that women who rated themselves lower on physical attractiveness used more makeup [25].

Although individual differences in physical facial attractiveness are larger than intra-individual differences caused by facial cosmetics [26], makeup is used to improve evaluation by others [6, 27] and may enhance prosocial feelings [25]. However, contrasting results were found regarding self-esteem and their relationship with makeup usage. Robertson and colleagues [25] found that self-esteem is negatively related to cosmetic usage whereas Al-Samydai et al. [28] findings pointed to a positive association. Nevertheless, the contrasting results may be due to the characteristics of the samples: the first study was conducted on 30 British undergraduate women and the latter on 606 Jordanian women. Studies investigating the association between self-esteem and makeup usage in different sociocultural settings are needed.

There are several ways to measure appearance, including perception of physical attractiveness (e.g., facial or body attractiveness rated by others), morphological measures (e.g., muscularity, body shape), and body image (e.g., individuals' attitude toward appearance). Body image is broadly used to measure appearance because it is the reflection of individual satisfaction with their physical appearance and the importance placed on always looking good [28]. Whereas appearance orientation measures the importance attributed to their own appearance, such as how important they think it is to always look good, appearance evaluation measures

how attractive the individual considers themselves, how satisfied they are with their own body [28]. Thus, body image measures self-perception of their attractiveness and also the pressure they put on themselves to always look good. Researchers relating makeup and body image tend to approach only the appearance evaluation factor, leaving out appearance orientation [e.g. 6, 29], despite several studies showing the impact social pressure exerts on individuals' self-perception [7], self-esteem [10], and even well-being [11].

Thus, our goal was to test if makeup usage in women (frequency of makeup usage, money spent on makeup, and time spent applying makeup per day) is predicted by general and social self-esteems, and body image (self-perceived attractiveness and importance one gives to tidiness). Despite some contrasting findings, Al-Samydai et al. [28] pointed out that makeup enhances women's social interactions and performance and Robertson et al. [25] found a positive relationship between makeup usage and self-presentation and self-consciousness. Therefore, we expected that makeup usage would be predicted by social-related aspects (appearance orientation and social self-esteem) rather than by appearance evaluation and general self-esteem.

## Materials and methods

### Participants

A total of 1,651 Brazilian women took part in the research. For the present study, 1,483 women between 18 and 75 years old ($M_{age}$ = 31.08; $SD$ = 11.15) entered in the final analyses. All the 168 participants younger than 18 years had their data excluded. This sample was comprised 32.2% of women with graduate degrees, 26.1% undergraduate students, 20.6% women with completed undergraduate degrees, 10.0% women with secondary education, 9.3% graduate students, and 1.7% with no education or unfinished secondary education. Most of the women considered themselves White (73.0%), 17.8% indicated mixed ethnicity (*pardo*), 4.8% identified as Black, 2.8% as Asian, and 1.6% indicated "Other" ethnicity.

About family income, most of the participants (27.4%) declared from approximately US$ 499 to 998 (exchange rate of the day December 28 2021), 19.4% declared from US$ 998 to 1,496, 15.4% declared from US$ 166 to 498, 15% declared more than US$ 2,494, 11.8% declared from US$ 1,497 to 1,995, 8% from US$ 1,996 to 2,493, and 3.1% up to US$ 165. The average Brazilian income per capita was US$ 313 in 2018 (when most of the data was collected) [30]. Most participants were from Southeast Brazil.

### Instruments

Participants completed a Qualtrics online questionnaire (Qualtrics, Provo, UT), containing sociodemographic questions, Cosmetics Use Inventory and additional questions, Social self-esteem questionnaire, General self-esteem questionnaire, and Body image scale.

**Sociodemographic questions.** This section included questions such as age, sex, gender, sexual orientation, ethnicity, relationship status, current pregnancy, and current socioeconomic status (educational level and family income).

**Cosmetics use inventory [31].** We used a part of an adapted version of the inventory [32] in which participants rate on a 7-point-scale the frequency they use from "never" to "always": 1. base, concealer, and/or powder; 2. mascara; 3. eyeliner or eye pencil; 4. shade; and 5. lipstick and/or gloss. Higher averaged scores correspond to higher levels of facial cosmetic use. We used the version translated (and back translated) into Brazilian Portuguese. All the variables were positively associated (i.e., women who use more one type of makeup tend to use the other types of makeup more frequently, too) (See S1 Table in S1 File).

Further, participants responded about their monthly expenses with makeup using the following options (in our survey in Brazilian Reals): USD 0, up to USD 2.50, USD 2.50–6, USD 6–10, USD 10–15, USD 15–20, USD 20–25, USD 25–50, more than USD 50. Time spent applying makeup per day was responded using the following options: less than 5 minutes, 5–10 minutes, 10–20 minutes, 20–30 minutes, more than 30 minutes.

**Social self-esteem questionnaire [9].** This is a 30-item instrument composed of phrases describing one's ability to deal with different social situations. The participant answers how accurately each sentence describes what her behavior or feelings would be in each situation on a six-point Likert scale. The higher the score, the greater the participant's ease in dealing with social situations (e.g., "I make friends easily"). We used the version translated (and back translated) into Brazilian Portuguese (Cronbach α = 0.95).

**General self-esteem questionnaire [8].** This is a 10-item instrument with a four-point Likert response scale. It contains affirmations about individuals' feelings and beliefs about themselves (e.g., "On the whole, I am satisfied with myself"). We used the version translated and adapted into Portuguese and validated for the Brazilian population (Cronbach α = 0.91) [33].

**Body image scale [28].** This is an attitudinal body image instrument composed of two subscales measuring appearance evaluation and appearance orientation with a five-point Likert response scale. The subscales are composed of 17 statements, 11 of them related to appearance orientation (e.g. "It is important that I always look good") and six related to appearance evaluation (e.g. "I like my looks just the way they are"). We used the version translated (and back translated) into Brazilian Portuguese (Cronbach α = 0.82).

## Procedure

After written ethical approval by the local Institutional Review Board of Anhembi Morumbi University (nr. 2.960.684), participants were recruited through social media and institutional e-mails. Thus, it was a sample based on convenience, and does not represent the Brazilian population. Participants completed informed consent and then responded to anonymous online questionnaires. Inclusion criteria were to have access to the Internet and to be a Brazilian woman 18 years old or older. Participants took 30 minutes on average to complete the survey.

## Data analyses

First, using IBM SPSS Statistics for Windows, version 21 (IBM Corp., Armonk, N.Y., USA), we checked data normality (See S2 Table in S1 File). Most data were not normally distributed, and we thus conducted exploratory non-parametric correlations among makeup usage, social and general self-esteems, and body image in order to verify correlations among the independent variables and test for multicollinearity. The independent variables were weakly and moderately associated, with low risk of multicollinearity (VIF ranged from 1.002 to 2.002).

Second, to test for a possible effect on makeup usage, social and general self-esteems, and body image entered as independent variables into categorical regressions (CATREG). We chose to use this analysis because it uses an optimal scaling feature that solves the problem of lack of linearity of the scales and it calculates an optimal regression equation and the effect of each independent variable (appearance orientation, appearance evaluation, general self-esteem, and social self-esteem) on the dependent variables (frequency of makeup usage, money spent on makeup, and time spent doing makeup per day). All statistical tests were performed with the significance level indicated at .05.

## Results

### Makeup usage descriptives

Most participants use makeup half of the time (26.2%) or sometimes (24.9%) and 44.9% spend less than five minutes applying makeup per day. Also, 19.6% spent nothing and 19.6% spent up to USD 2.50 on makeup per month. See S3 to S5 Tables in S1 File for detailed data.

### Correlations between makeup usage and social and general self-esteem, and body image attitudes

Kendall correlation indicated that money spent on makeup per month, time spent applying makeup per day, and frequency of makeup usage are moderately and positively correlated (See Table 1). Further, these three measures of makeup usage are moderately and positively correlated to appearance orientation. Money spent on makeup per month and frequency of makeup usage are weakly and positively related to social self-esteem. Frequency of makeup usage also presented a positive and weak correlation with general self-esteem. Social self-esteem was moderately and positively correlated with general self-esteem, weakly and positively associated with appearance evaluation and appearance orientation. General self-esteem was moderately and positively associated with appearance evaluation.

The sociodemographic variables (age, family income, and educational level) were weakly and positively associated with money spent on makeup, frequency of makeup usage, general self-esteem, social self-esteem, and appearance evaluation. There were no associations among the sociodemographic variables and time spent on makeup and appearance orientation.

### The effect of general and social self-esteems and body image on makeup usage

To test for a possible effect of social and personal self-esteems, and body image on makeup usage, we conducted three categorical regression models, with money spent on makeup per month, time spent applying makeup per day, and frequency of makeup usage as dependent variables. We also included the sociodemographic variables age, family income, and educational level in the analyses in order to control the variability of our sample (Table 2).

**Table 1. Kendall correlations between makeup usage, social self-esteem, general self-esteem, body image, and sociodemographic variables.**

|  | 1 | 2 | 3 | 4 | 5 | 6 | 7 | 8 | 9 |
|---|---|---|---|---|---|---|---|---|---|
| 1. Money spent on makeup per month |  |  |  |  |  |  |  |  |  |
| 2. Time spent on makeup per day | .467** |  |  |  |  |  |  |  |  |
| 3. Frequency of makeup usage | .428** | .488** |  |  |  |  |  |  |  |
| 4. General self-esteem | 0,033 | -0.003 | .053** |  |  |  |  |  |  |
| 5. Appearance orientation | .290** | .371** | .351** | .011 |  |  |  |  |  |
| 6. Appearance evaluation | -.024 | .018 | .039 | .369** | .027 |  |  |  |  |
| 7. Social Self-esteem | .090** | .03 | .085** | .430** | .078** | .270** |  |  |  |
| 8. Educational level | .075** | -.027 | .097** | .194** | .004 | .101** | .148** |  |  |
| 9. Family income | .088** | -.035 | .054** | .162** | .029 | .064** | .154** | .243** |  |
| 10. Age | .135** | .025 | .123** | .213** | -.009 | .048* | .177** | .431** | .116** |

** Correlation is significant at the 0.01 level (2 ends).

* The correlation is significant at the 0.05 level (2 ends).

**Table 2. Categorical regressions.**

|  | R | Adjusted R$^2$ | F | df | p | Factor | B | p |
|---|---|---|---|---|---|---|---|---|
| **Frequency of makeup usage** | .536 | .274 | 22.891 | 984 | ≤.001 |  |  |  |
|  |  |  |  |  |  | General **self-esteem** | -.083 | .155 |
|  |  |  |  |  |  | **Social self-esteem** | .0331 | .624 |
|  |  |  |  |  |  | **Appearance orientation** | .487 | ≤.001 |
|  |  |  |  |  |  | **Appearance evaluation** | .048 | .386 |
|  |  |  |  |  |  | Age | .181 | ≤.001 |
|  |  |  |  |  |  | Family income | .052 | .065 |
|  |  |  |  |  |  | Educational level | .019 | .704 |
| **Money spent on makeup** | .461 | .195 | 12.152 | 1014 | ≤.001 |  |  |  |
|  |  |  |  |  |  | General **self-esteem** | -.084 | .009 |
|  |  |  |  |  |  | **Social self-esteem** | .087 | ≤.001 |
|  |  |  |  |  |  | **Appearance orientation** | .393 | ≤.001 |
|  |  |  |  |  |  | **Appearance evaluation** | -.073 | .013 |
|  |  |  |  |  |  | Age | 0.195 | ≤.001 |
|  |  |  |  |  |  | Family income | 0.115 | ≤.001 |
|  |  |  |  |  |  | Educational level | -.078 | .380 |
| **Time spent on makeup** | .477 | .214 | 17.266 | 17, 997 | ≤.001 |  |  |  |
|  |  |  |  |  |  | General **self-esteem** | -.119 | .119 |
|  |  |  |  |  |  | **Social self-esteem** | .021 | .873 |
|  |  |  |  |  |  | **Appearance orientation** | .455 | ≤.001 |
|  |  |  |  |  |  | **Appearance evaluation** | .054 | .295 |
|  |  |  |  |  |  | Age | .074 | .011 |
|  |  |  |  |  |  | Family income | -.044 | .205 |
|  |  |  |  |  |  | Educational level | -.078 | .315 |

Only appearance orientation and age predicted time spent applying makeup per day and frequency of makeup usage, whereas all variables except educational level predicted money spent on makeup.

## Discussion

The aim of the study was to verify if general and social self-esteems and body image (i.e., appearance orientation and appearance evaluation) were associated with makeup usage among Brazilian women. Altogether, our findings suggested that women who feel comfortable with their appearance and have higher general self-esteem spent less money on makeup whereas women with higher social self-esteem spent more money on makeup; and women who allocated more importance to the way they looked not only spent more money on makeup but spent more time applying makeup and using makeup more frequently.

Appearance orientation was a significant predictor of makeup usage. Thus, women who give more importance to their appearance and are always neat, use makeup more often, spend more time applying makeup, and spend more money on makeup. Similarly, Robertson et al. [25] found a positive relationship between cosmetic usage and self-presentation. In women, appearance orientation is also linked to neuroticism and narcissism [34], eating disorders [35], and drive for muscularity in men [36]. Women frequently have their bodies objectified, i.e. treated like an object that exists to please others. Through self-objectification, they disconnect their bodies from their persons, and sometimes internalize this perspective and start evaluating and treating themselves as mere bodies [37], highlighting the visual assessment. Women with

higher levels of self-objectification would place more attention to their appearance and grooming instead of other aspects, such as identity development. Therefore, appearance orientation may be used as a measure of self-objectification [34]. A deeper investigation about appearance orientation and self-objectification should be conducted in order to clarify if they are in fact measuring similar traits.

Further, when adolescents with positive body image were interviewed, they claimed their family and friends used to talk about their appearance, but not about their bodies (i.e., they comment about their clothing, hair style, makeup, etc., but not about their physical traits, such as how fat they are [38]). Thus, someone's appearance orientation is more liable in commentaries than someone's body. It suggests that appearance orientation would be more susceptible to social influences than appearance evaluation, and consequently, would be positively related to appearance modifications, including makeup usage. This would also explain why accepting oneself is negatively related to purchasing makeup.

Corroborating Frisén and Holmqvist's [38] results, we found that women with higher appearance evaluation, i.e. women who were more satisfied with their appearance, tended to spend less money on makeup. Our study supported findings of Robertson and colleagues [25] who reported an inverse association between cosmetic usage and self-rated physical appearance. For these individuals, makeup usage may not be related to satisfaction with their body, so it does not make them feel physically more attractive. Indeed, cosmetics have a smaller impact on individuals higher in attractiveness than on less attractive individuals [26].

Frederick and Reynolds [39] presented the cognitive behavioral model in which makeup would be an appearance fixing strategy, being a response to emotions and thoughts related to body image. That is, factors experienced throughout an individual's life influenced one's body image by associating their appearance schema. Future experiences can activate this model, influencing how this new information is processed. The thoughts and emotions related to one's schema will respond by adjusting self-regulatory processes. Makeup usage, thus, would be a way to improve body image through fixing imperfections in people who are not satisfied with their appearance (appearance schema) and are concerned about it.

Additionally, Mafra et al. [40] conducted a study on Brazilian men and women with low socioeconomic status and found that spending more money on cosmetics did not make women feel better about themselves (e.g., more attractive). According to a review by Tylka and Wood-Barcalow [41], positive body image is the acceptance of one's own body, feeling happy and complete even knowing its imperfections and that it is not consistent with idealized images. Nevertheless, others' perceptions also may influence individuals' positive body image [39] as well as the type of watched advertisement may influence women's self-esteem, body image, and mood [42].

Social self-esteem is a positive predictor of money spent on makeup per month. Although Robertson et al. [25] found a negative association between cosmetic usage and social confidence, adolescent girls reported to use makeup because they wanted to feel admired by the public [23], suggesting that makeup functions improve social impressions. In a recent review, Davis and Arnocky [18] argued that makeup may be used as a strategy to enhance social status. In fact, besides bringing advantages in attracting mates and competing with rivals [21], makeup usage was positively associated with social interaction and performance [43], with women who use makeup being perceived as more competent [44], more dominant, and higher in social prestige [45]. As social interactions are important for people with high social self-esteem levels, makeup may be a tool to increase confidence in interpersonal relationships. On the other hand, women who feel good about themselves (i.e., with high general self-esteem) tended to spend less money on makeup usage. This result also corroborated Robertson et al.

[25], in which a negative correlation between makeup usage and general self-esteem was found.

According to a recent study, makeup use also affects other women. After being exposed to pictures of same-sex peers wearing makeup, Australian undergraduate women reported willingness to change their appearance, for example, hair, skin [29]. Possibly, women who give more importance to physical appearance invest more money also on other beauty products, not just makeup. Future research investigating how other products that can enhance women's appearance relate to appearance orientation and ideal stereotypes of beauty could make a great contribution to the field.

Overall, our study suggested that women with greater self-esteem associated with social interactions would attribute higher importance to their appearance, resulting in more makeup usage. Makeup usage may enhance women's confidence to deal with social situations. However, our study was cross-sectional, thus an experimental study could complement our approach by testing if social-related aspects influence the consumption of makeup more than intrinsic-related aspects. Another important point to highlight is that the social influence on makeup usage is rather speculative since we have not directly asked the participants about the possible impact of the media and social networks on their body image. Finally, our sample was composed by a majority of highly educated Caucasian women of medium to high family incomes, which does not fully represent the Brazilian population (mostly composed by Black and *mixed* ethnicities in low educational levels and low family income earnings). A study trying to reach more women outside the university may better represent the Brazilian population.

## Supporting information

**S1 File. Additional analyses.**
(DOCX)

## Acknowledgments

We are grateful for English proofreading and helpful suggestions made by Prof. Daniel J. Kruger. We are also grateful to the participants and researchers who donated their time for this to become a reality.

## Author Contributions

**Conceptualization:** Marco A. C. Varella, Jaroslava V. Valentova.

**Data curation:** Anthonieta Looman Mafra, Caio S. A. Silva.

**Formal analysis:** Anthonieta Looman Mafra.

**Funding acquisition:** Anthonieta Looman Mafra, Caio S. A. Silva, Marco A. C. Varella, Jaroslava V. Valentova.

**Investigation:** Anthonieta Looman Mafra.

**Methodology:** Marco A. C. Varella, Jaroslava V. Valentova.

**Project administration:** Anthonieta Looman Mafra, Jaroslava V. Valentova.

**Resources:** Jaroslava V. Valentova.

**Supervision:** Jaroslava V. Valentova.

**Visualization:** Anthonieta Looman Mafra.

**Writing – original draft:** Anthonieta Looman Mafra.

**Writing – review & editing:** Anthonieta Looman Mafra, Caio S. A. Silva, Marco A. C. Varella, Jaroslava V. Valentova.

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
