## [Decision Letter · Decision Letter 0]

9 Dec 2021

PONE-D-21-23860The contrasting effects of body image and self-esteem in the makeup usagePLOS ONE

Dear Dr. Mafra,

Thank you for submitting your manuscript to PLOS ONE. After careful consideration, we feel that it has merit but does not fully meet PLOS ONE’s publication criteria as it currently stands. Therefore, we invite you to submit a revised version of the manuscript that addresses the points raised during the review process. Please submit your revised manuscript by Jan 23 2022 11:59PM. If you will need more time than this to complete your revisions, please reply to this message or contact the journal office at plosone@plos.org. Please include the following items when submitting your revised manuscript:A rebuttal letter that responds to each point raised by the academic editor and reviewer(s). You should upload this letter as a separate file labeled 'Response to Reviewers'.A marked-up copy of your manuscript that highlights changes made to the original version. You should upload this as a separate file labeled 'Revised Manuscript with Track Changes'.An unmarked version of your revised paper without tracked changes. You should upload this as a separate file labeled 'Manuscript'.If applicable, we recommend that you deposit your laboratory protocols in protocols.io to enhance the reproducibility of your results. Protocols.io assigns your protocol its own identifier (DOI) so that it can be cited independently in the future. For instructions see: https://journals.plos.org/plosone/s/submission-guidelines#loc-laboratory-protocols. Additionally, PLOS ONE offers an option for publishing peer-reviewed Lab Protocol articles, which describe protocols hosted on protocols.io. Read more information on sharing protocols at https://plos.org/protocols?utm_medium=editorial-email&utm_source=authorletters&utm_campaign=protocols.

We look forward to receiving your revised manuscript.

Kind regards,

Piotr Sorokowski

Academic Editor

PLOS ONE

Journal Requirements:

Reviewers' comments:

Reviewer's Responses to Questions

**Comments to the Author**

1. Is the manuscript technically sound, and do the data support the conclusions?

Reviewer #1: Yes

Reviewer #2: Yes

2. Has the statistical analysis been performed appropriately and rigorously? 

Reviewer #1: Yes

Reviewer #2: Yes

3. Have the authors made all data underlying the findings in their manuscript fully available?

Reviewer #1: Yes

Reviewer #2: Yes

4. Is the manuscript presented in an intelligible fashion and written in standard English?

Reviewer #1: Yes

Reviewer #2: Yes

5. Review Comments to the Author

Reviewer #1: Dear Editor, Dear Authors,

I would like to thank for the opportunity to review this interesting manuscript. It reports on relationships between self-esteem, body image, and makeup usage. The study was conducted on a satisfactory large sample (N = 1483 Brazilian women). The strength of the paper is that it adds to a heated discussion on self-presentation modification, which recently drew wide public attention. For this reason, I believe that the paper would be appealing to many of the PLOS ONE Readers. However, the manuscript has several issues, which I outline below.

Major issues:

1. Participants' paragraph. In my opinion, a few things are missing:

- A description of the exclusion criteria (why 168 participants were excluded from further analyses?).

- Age range (what age was the oldest participants?).

- Details on participants' economic situation. The sample was somewhat not representative of a Brazilian population (it mainly consisted of "highly educated Caucasia women"). Thus, I am concerned that the sample was also rather on the upper end of the socio-economic ladder, and thus, it should be highlighted throughout the ms that the conclusions from the study are limited.

- For the above reasons, it would also be interesting to know how the participants were recruited.

2. "Time spent applying makeup per day was responded using the following options: less than 5 minutes, 5-10 minutes, 10-20 minutes, 20-30 minutes, more than 30 minutes." I am wondering why the Authors chose these categories. In the first sentence of this manuscript, the Authors refer that 78% of American women spend 1 hour a day on their appearance. Wasn't the ceiling reached in the current study?

3. "Most data were not normally distributed" What were the skewness and kurtosis values? The Authors may consider adding this information (e.g., in the supplementary material).

4. "Makeup usage descriptives" I was wondering whether the Authors could add a more detailed table with information on all categories.

5. The statistical analyses. The Authors did not control for the economic status nor age in the analyses. I wonder what the results are if running these extended models.

Minor issues:

1. I would suggest softening a final sentence in the abstract–"The results suggest that the significance given to appearance and social interactions have an important effect on makeup usage in women." Having an effect on something implies that A affects B, while the present survey was observational in nature.

2. The frequency of cosmetics use inventory. I am wondering over the rationale for testing Cronbach's alpha of 5 questions about the frequency of using the given types of cosmetics. I imagine a situation where a given woman wears base, mascara, and eyeliner every day, but not lipstick and shade. She is, thus, frequently using a given set of cosmetics, artificially lowering the reliability coefficient. As a metaphor, someone can wear red socks every day but not blue ones. Therefore, this person frequently wears socks, but if we asked them two questions, whether they wear red socks and blue socks, their Cronbach's alpha would be unsatisfactory.

3. First two sentences from the introduction. I would encourage the Authors to elaborate on these findings. The claims made are influential and require strong scrutiny in presenting them. The provided source for this sentence actually goes back to a study from 2014 on a nationally representative sample of over 2k online American adults.

Furthermore, the second sentence goes "They spent 10 minutes on makeup (…).". However, the Authors cite another study, and thus, I would consider rephrasing (e.g., something like "Another study showed...").

Also, the Authors described two studies on American women and then wrote: "these examples illustrate the importance women in Western societies attribute to physical appearance and self-care". Although American women undoubtedly are an example of Western women, I suggest adding at least one more example of a Western population or rephrasing the beginning.

4. "Social self-esteem is positively linked to use of social network sites [10,11]. Thus, the increase in digital media usage has also increased the pressure to look as good as possible because people are increasingly exposed to images and videos of perfectly looking individuals [12]." This line of thought is worth pursuing. However, the Authors might think of first introducing why social self-esteem is relevant to the use of social network sites.

5. "Interestingly, 37% of the interviewed women reported they started using makeup because they were unsatisfied with their appearance [12]." I would suggest the Authors add a brief description of the given population when referring to a given study (throughout the manuscript).

6. Linguistic remark. The Authors may consider using the tense consistently throughout the ms (i.e., past simple or present, when referring to other studies' results). Now it is sometimes present simple, sometimes past simple.

7. "Although individual differences in physical facial attractiveness are larger than intra individual differences caused by facial cosmetics [22], makeup is used to enhance not only self-esteem and self-perceptions, but also perception by others [6,23] and may enhance prosocial feelings [21]". I would suggest the Authors rephrase this sentence as it reads a bit odd.

8. "However, contrasting results were found regarding self-esteem and their relationship with makeup usage. Robertson and colleagues [21] found that self-esteem is negatively related to cosmetic usage whereas Al-Samydai et al. [24] findings pointed to a 76 positive association." These contrasting results may stem from the fact that Robertson et al. (2008) study was conducted on an extremely small sample – 30 British undergraduates, while Al-Samydai study was conducted on 606 Jordanian women.

9. "There are several ways to measure appearance, including physical attractiveness per se, and body image." The Authors may clarify what exactly they refer to (e.g., (…) to measure the attitudes toward appearance).

10. "Thus, our general goal is to test if makeup usage in women" For brevity reasons, the Authors may consider omitting 'general.'

11. "This sample was composed by 32.2% of (…)" The Authors may rephrase "This sample was composed by 32.2% of women "into "This sample comprised 32.2% of women (…)".

12. "The independent variables are were weakly and moderately associated, with low risk of multicollinearity" Please, correct this sentence. Furthermore, what were the variance inflation factors?

13. ", Also, 19.6% spend nothing and 19.6% spend up to USD 2.50 on makeup per month (…)" I believe there is a comma instead of a dot.

14. Table 1. Could the Authors also add relationships between the variables of interest and age, economic status, and educational status into the Table?

I would also suggest adding asterisks to mark significant relationships while removing the p values (it would make the Table more readable).

15. I suggest the Authors unify the language used to describe self-esteem. Sometimes it is 'personal self-esteem' (Table 2), while in Table 1, it is "general self-esteem.

16. The Authors may also consider discussing the claims that physical appearance (and enhancing it) may serve as a female strategy to increase social status (for a review, see Davis & Arnocky, 2020). https://doi.org/10.1007/s10508-020-01745-4

Kind regards

Reviewer #2: This is an interesting study and the authors have collected a unique dataset using cutting edge methodology. The paper is generally well written and structure However, in my opinion this is a good paper

6. PLOS authors have the option to publish the peer review history of their article (what does this mean?). If published, this will include your full peer review and any attached files.

Reviewer #1: No

Reviewer #2: No

---

## [Author Response · Author response to Decision Letter 0]

17 Feb 2022

Review Comments to the Author

Reviewer #1: Dear Editor, Dear Authors,

I would like to thank for the opportunity to review this interesting manuscript. It reports on relationships between self-esteem, body image, and makeup usage. The study was conducted on a satisfactory large sample (N = 1483 Brazilian women). The strength of the paper is that it adds to a heated discussion on self-presentation modification, which recently drew wide public attention. For this reason, I believe that the paper would be appealing to many of the PLOS ONE Readers. However, the manuscript has several issues, which I outline below.

Response: We thank the reviewer for recognizing the strengths of our study and for the valuable suggestions. The modifications made were essential for improving the manuscript. 

Major issues:

1. Participants' paragraph. In my opinion, a few things are missing:

- A description of the exclusion criteria (why 168 participants were excluded from further analyses?).

Response: The excluded participants were younger than 18 years. Although the consent term had specified the minimum age to participate in the study, some women under 18 years answered the questionnaire. We added this information to the ms. Now it reads: “All the 168 participants younger than 18 years had their data excluded.”

2. Age range (what age was the oldest participants?).

Response: Thank you for noticing. We had described only the mean age. The oldest participant was 75 years old. We added the information into the description of the sample. Now it reads: “A total of 1,651 Brazilian women took part in the research. For the present study, 1,483 women between18 and 75 years old (Mage = 31.08; SD = 11.15) entered in the final analyses.”

3. Details on participants' economic situation. The sample was somewhat not representative of a Brazilian population (it mainly consisted of "highly educated Caucasia women"). Thus, I am concerned that the sample was also rather on the upper end of the socio-economic ladder, and thus, it should be highlighted throughout the ms that the conclusions from the study are limited.

Response: Agreed with the reviewer. We added the following part in the end of the “Participants” topic: “About family income, most of the participants’ families (27.4%) earned approximately between US$ 499 and 998, 19.4% earned between US$ 998 and 1,496, 15.4% earned between US$ 166 and 498, 15% earned more than US$ 2,494, 11.8% earned from US$ 1,497 and 1,995, 8% earned between US$ 1,996 and 2,493, and 3.1% earned up to US$ 165 (exchange rate of the day December 28 2021). The average Brazilian income per capita was US$ 313 in 2018 (when most of the data was collected)”. As the reviewer had noticed, the paper sample does not represent the average Brazilian population. So we added this as a study limitation into the Discussion, and now it reads as follows: “our sample was composed by a majority of highly educated White women of medium to high family incomes, which does not fully represent the Brazilian population (mostly composed by Black and mixed ethnicities with low educational levels and low family income).”

4. For the above reasons, it would also be interesting to know how the participants were recruited.

Response: We included this information in the “Procedure” section on page 8 “participants were recruited through social media and institutional e-mails. Thus, it was a sample based on convenience, and does not represent the Brazilian population.”

5. "Time spent applying makeup per day was responded using the following options: less than 5 minutes, 5-10 minutes, 10-20 minutes, 20-30 minutes, more than 30 minutes." I am wondering why the Authors chose these categories. In the first sentence of this manuscript, the Authors refer that 78% of American women spend 1 hour a day on their appearance. Wasn't the ceiling reached in the current study?

Response: Thank you for noticing that the text may confuse the reader. We failed to distinguish the terms. The “time spent on appearance” can include a broader range of other behaviors besides makeup application, such as hair care, skin care or dressing up. Based on that, we considered that women should spend less than one hour applying daily makeup, as this behavior would be only a part of the total time spent on appearance. We added information in the introduction. Now it reads as follows: “Surveys with American women showed that 78% spend one hour per day on their appearance (e.g., hair treatments, dressing up, and makeup).” 

6. "Most data were not normally distributed" What were the skewness and kurtosis values? The Authors may consider adding this information (e.g., in the supplementary material).

Response: We added the descriptive information, including skewness and kurtosis values, into the Supporting information. Please refer to Table S2.

7. "Makeup usage descriptives" I was wondering whether the Authors could add a more detailed table with information on all categories.

Response: We provided further details in the Supporting information (S3 to S5 Table).

8. The statistical analyses. The Authors did not control for the economic status nor age in the analyses. I wonder what the results are if running these extended models.

Response: We agree that these are important variables to include to control the results. We ran the analyses again, inserting the variables age, family income, and level of education into the final regression model. The results were virtually the same but level of education was not a predictor of the dependent variables and age was a predictor of all dependent variables. We modified the result section. Please check “The effect of general and social self-esteems and body image on makeup usage” in the Results section.

Minor issues:

1. I would suggest softening a final sentence in the abstract–"The results suggest that the significance given to appearance and social interactions have an important effect on makeup usage in women." Having an effect on something implies that A affects B, while the present survey was observational in nature.

Response: We agree, and we rephrased the sentence, as suggested. Now it reads as follows: “The results suggest that the significance given to appearance and social interactions are importantly associated with makeup usage in women.”

2. The frequency of cosmetics use inventory. I am wondering over the rationale for testing Cronbach's alpha of 5 questions about the frequency of using the given types of cosmetics. I imagine a situation where a given woman wears base, mascara, and eyeliner every day, but not lipstick and shade. She is, thus, frequently using a given set of cosmetics, artificially lowering the reliability coefficient. As a metaphor, someone can wear red socks every day but not blue ones. Therefore, this person frequently wears socks, but if we asked them two questions, whether they wear red socks and blue socks, their Cronbach's alpha would be unsatisfactory.

Response: We agree with the reviewer. For this reason, we decided to delete the Cronbach alpha for this scale. Instead, we added a correlation table among the different cosmetic types usage into the Supporting information and also added this information in the Methods. It now reads as follows: “All the variables were positively associated (i.e., women who use more one type of makeup tend to use the other types of makeup more frequently, too) (See Table S1 in the Supporting information).”

3. a) First two sentences from the introduction. I would encourage the Authors to elaborate on these findings. The claims made are influential and require strong scrutiny in presenting them. The provided source for this sentence actually goes back to a study from 2014 on a nationally representative sample of over 2k online American adults.

Response: We agree that there was more data that could have been addressed in the introduction and describe more the importance of appearance for American women. The first paragraph of the introduction reads now as follows: “Surveys with American women showed that 78% spent one hour per day on their appearance (e.g., hair treatments, dressing up, and makeup). Taking on average 55 minutes of women’s day, hair and makeup seems to need more time invested than other appearance related behaviors. [1] American women did not only spend time on active appearance enhancing behaviors, but between the most watched categories by women on YouTube, the top two are appearance related [1]. Another study showed that American women spent, on average, 10 minutes on makeup in the morning and 85% tended to apply at least 16 products on their faces before leaving home. The results also concluded that New York women spent around 300,000 US dollars during their lifetime on facial cosmetics [2]. These examples illustrate the importance American women attribute to physical appearance and self-care [e.g., 3].”

b) Furthermore, the second sentence goes "They spent 10 minutes on makeup (…).". However, the Authors cite another study, and thus, I would consider rephrasing (e.g., something like "Another study showed...").

Response: We agree it is more appropriate and corrected the phrase as suggested, see our reply above. 

c) Also, the Authors described two studies on American women and then wrote: "these examples illustrate the importance women in Western societies attribute to physical appearance and self-care". Although American women undoubtedly are an example of Western women, I suggest adding at least one more example of a Western population or rephrasing the beginning.

Response: American women may not represent all Western societies. We agree that it is wiser to change the statement. It reads now as follows: “these examples illustrate the im portance American women attribute to physical appearance and self-care”

4. "Social self-esteem is positively linked to use of social network sites [10,11]. Thus, the increase in digital media usage has also increased the pressure to look as good as possible because people are increasingly exposed to images and videos of perfectly looking individuals [12]." This line of thought is worth pursuing. However, the Authors might think of first introducing why social self-esteem is relevant to the use of social network sites.

Response: We reformulated the explanation to make the distinction between social and general self-esteem clear. Now it reads as follows: “Social self-esteem is positively linked to use of social network sites [10,11] whereas general self-esteem appears to be inversely proportional to social network sites [12, 13], especially in women [14]. Vogel and colleagues [15] found that social comparison negatively affected general self-esteem. Thus, the increase in digital media usage has also increased the pressure to look as good as possible because people were increasingly exposed to images and videos of very good looking individuals [16]. On the other hand, Steinsbekk and colleagues [14] found that self-oriented social network sites use was not related to general self-esteem. Therefore, social network sites might increase social self-esteem by promoting more social interactions”

5. "Interestingly, 37% of the interviewed women reported they started using makeup because they were unsatisfied with their appearance [12]." I would suggest the Authors add a brief description of the given population when referring to a given study (throughout the manuscript).

Response: We agree that it is important to describe the study sample in cases like this. We added the information and now it reads: “Interestingly, Kelley [12] interviewed 132 American college women and found that 37% of them reported they started using makeup because they were unsatisfied with their appearance”.

We also tried to accomplish this suggestion with other studies in the manuscript. 

6. Linguistic remark. The Authors may consider using the tense consistently throughout the ms (i.e., past simple or present, when referring to other studies' results). Now it is sometimes present simple, sometimes past simple.

Response: We reviewed the entire manuscript and used past simple. Thank you for noticing.

7. "Although individual differences in physical facial attractiveness are larger than intra individual differences caused by facial cosmetics [22], makeup is used to enhance not only self-esteem and self-perceptions, but also perception by others [6,23] and may enhance prosocial feelings [21]". I would suggest the Authors rephrase this sentence as it reads a bit odd.

Response: Agreed, it was confusing. We tried to fix the sentence, and now it reads as follows: “ Although individual differences in physical facial attractiveness are larger than intra-individual differences caused by facial cosmetics [22], makeup is used to improve evaluation by others [6,23] and may enhance prosocial feelings [21].”

8. "However, contrasting results were found regarding self-esteem and their relationship with makeup usage. Robertson and colleagues [21] found that self-esteem is negatively related to cosmetic usage whereas Al-Samydai et al. [24] findings pointed to a 76 positive association." These contrasting results may stem from the fact that Robertson et al. (2008) study was conducted on an extremely small sample – 30 British undergraduates, while Al-Samydai study was conducted on 606 Jordanian women.

Response: Thank you for this notion. We added this possible explanation into the results section. It reads now as follows: “Nevertheless, the contrasting results may be due to the characteristics of the samples: the first study was conducted on 30 British undergraduate women and the latter on 606 Jordanian women. Studies investigating the association between self-esteem and makeup usage in different sociocultural settings are needed.”

9. "There are several ways to measure appearance, including physical attractiveness per se, and body image." The Authors may clarify what exactly they refer to (e.g., (…) to measure the attitudes toward appearance).

Response: We agree with the reviewer. It was not clear in the phrase what exactly we were measuring, so we tried to fix it. It reads now: “There are several ways to measure appearance, including perception of physical attractiveness (e.g., facial or body attractiveness rated by others), morphological measures (e.g., muscularity, body shape), and body image (e.g., individuals’ attitude toward appearance).”

10. "Thus, our general goal is to test if makeup usage in women" For brevity reasons, the Authors may consider omitting 'general.'

Response: We deleted the word “general” accordingly. 

11. "This sample was composed by 32.2% of (…)" The Authors may rephrase "This sample was composed by 32.2% of women" into "This sample comprised 32.2% of women (…)".

Response: We corrected the sentence as suggested. 

12. "The independent variables are were weakly and moderately associated, with low risk of multicollinearity" Please, correct this sentence. Furthermore, what were the variance inflation factors?

Response: Thank you for noticing the mistype. We corrected it. The VIF ranged from 1.002 to 2.002. We added this information to the manuscript. Now it reads: “The independent variables were weakly and moderately associated, with low risk of multicollinearity (VIF ranged from 1.002 to 2.002).”

13. ", Also, 19.6% spend nothing and 19.6% spend up to USD 2.50 on makeup per month (…)" I believe there is a comma instead of a dot.

Response: Thank you for spotting this mistake . We fixed it. 

14. Table 1. Could the Authors also add relationships between the variables of interest and age, economic status, and educational status into the Table?

I would also suggest adding asterisks to mark significant relationships while removing the p values (it would make the Table more readable).

Response: We added the suggested variables to the table and made the additional changes. Please refer to Table 1 to see the new table. It is now more readable.

We also added the following paragraph to the manuscript: “The sociodemographic variables (age, family income, and educational level) were weakly and positively associated with money spent on makeup, frequency of makeup usage, general self-esteem, social self-esteem, and appearance evaluation. There were no associations among the sociodemographic variables and time spent on makeup and appearance orientation.”

15. I suggest the Authors unify the language used to describe self-esteem. Sometimes it is 'personal self-esteem' (Table 2), while in Table 1, it is "general self-esteem.

Response: Thank you for noticing, we corrected it accordingly. It occurred due to the initial SPSS labels.

16. The Authors may also consider disc ussing the claims that physical appearance (and enhancing it) may serve as a female strategy to increase social status (for a review, see Davis & Arnocky, 2020). https://doi.org/10.1007/s10508-020-01745-4

Response: It is a great paper that we should have addressed before. Thank you for the suggestion. Now it reads as follows: “ In a recent review, Davis and Arnocky [44] argued that makeup may be used as a strategy to enhance social status. In fact, besides bringing advantages in attracting mates and competing with rivals [21], some studies showed that makeup usage was also positively associated with social interaction and performance [28], with women who use makeup being perceived as more competent [45], more dominant, and higher in social prestige [46]. As social interactions are important for people with high social self-esteem levels, makeup may be a tool to increase confidence in interpersonal relationships. On the other hand, women who feel good about themselves (i.e., with high general self-esteem) tended to spend less money on makeup usage. This result also corroborated Robertson et al. [25], in which a negative correlation between makeup usage and general self-esteem was found.”

---

## [Editor Report · Decision Letter 1]

28 Feb 2022

The contrasting effects of body image and self-esteem in the makeup usage

PONE-D-21-23860R1

Dear Dr. Mafra,

We’re pleased to inform you that your manuscript has been judged scientifically suitable for publication and will be formally accepted for publication once it meets all outstanding technical requirements.

Kind regards,

Piotr Sorokowski

Academic Editor

PLOS ONE

Additional Editor Comments (optional):

thanks for the corrections. Congratulations on your new publication
---

## [Editor Report · Acceptance letter]

17 Mar 2022

PONE-D-21-23860R1 

The contrasting effects of body image and self-esteem in the makeup usage 

Dear Dr. Mafra:

I'm pleased to inform you that your manuscript has been deemed suitable for publication in PLOS ONE. Congratulations! Your manuscript is now with our production department. 

Kind regards, 

on behalf of

Dr. Piotr Sorokowski 

Academic Editor

PLOS ONE